# Molecular and physiological characterization of *Fusarium* strains associated with different diseases in date palm

Amgad A. Saleh [1,2]*, Anwar H. Sharafaddin [1], Mahmoud H. El_Komy[1,3], Yasser E. Ibrahim [1,3], Younis K. Hamad[1,4]

1 Department of Plant Protection, College of Food and Agriculture Sciences, King Saud University, Riyadh, Saudi Arabia, 2 Agricultural Genetic Engineering Research Institute, Agriculture Research Center, Giza, Egypt, 3 Plant Pathology Institute, Agriculture Research Center, Giza, Egypt, 4 Plant Pathology Department, Faculty of Agriculture, Alexandria University, Alexandria, Egypt

* asa7976@gmail.com

**Data Availability Statement:** Supplementary files containing data sets of SSR genotyping, growth of *Fusarium* strains on different temperature and

## Abstract

Several species of *Fusarium* cause serious diseases in date palm worldwide. In the present work, 14 SSR markers were used to assess the genetic variation of *Fusarium* strains isolated from diseased trees in Saudi Arabia. We also studied the effect of different temperatures on mycelial growth of these strains. The pathogenicity of four strains of *F. proliferatum* was also evaluated on local date palm cultivars. Eleven SSR markers amplified a total of 57 scorable alleles from *Fusarium* strains. Phylogenetic analysis showed that *F. proliferatum* strains grouped in one clade with 95% bootstrap value. Within *F. proliferatum* clade, 14 SSR genotypes were identified, 9 of them were singleton. Four out of the five multi-individual SSR genotypes contained strains isolated from more than one location. Most *F. solani* strains grouped in one clade with 95% bootstrap value. Overall, the SSR markers previously developed for *F. verticillioides* and *F. oxysporum* were very useful in assessing the genetic diversity and confirming the identity of Saudi *Fusarium* strains. The results from the temperature study showed significant differences in mycelial growth of *Fusarium* strains at different temperatures tested. The highest average radial growth for *Fusarium* strains was observed at 25˚C, irrespective of species. The four *F. proliferatum* strains showed significant differences in their pathogenicity on date palm cultivars. It is anticipated that the assessment of genetic diversity, effect of temperature on hyphal growth and pathogenicity of potent pathogenic *Fusarium* strains recovered from date palm-growing locations in Saudi Arabia can help in effectively controlling these pathogens.

## Introduction

Date palm (*Phoenix dactylifera* L.) is one of the most important fruit crops in the arid climates including the Arabian Peninsula, North Africa and the Middle East regions [1]. The total world date palm production is around 9.08 million tons harvested from a total area of 1.38 million ha [2]. The kingdom of Saudi Arabia produces ca 17% of the total world production,

pathogenicity experiments are available at https://doi.org/10.6084/m9.figshare.14645202.v1.

**Funding:** AAS received a research fund from the Deanship of Scientific Research at King Saud University, Riyadh. The fund is under the research group no. RG-1440-001. The funders had no role in study design, data collection and analysis, decision to publish, or preparation of the manuscript.

**Competing interests:** The authors have declared that no competing interests exist.

harvested from 450 date palm cultivars [2]. Date palm is affected by several diseases that cause significant economic losses. Most of these diseases are associated with fungal pathogens [1]. *Fusarium* is considered the most important pathogenic fungal genus on date palms. The most destructive *Fusarium* species causing losses to date palm is *F. oxysporum* f. sp. *albedinis*, the causal agent of Bayoud disease [3]. The other pathogenic *Fusarium* species on date palms include *F. solani*, the causal agent of yellow death and sudden decline diseases [4] and *F. proliferatum*, the causal agent of inflorescence rot and bunch fading diseases [5]. In Saudi Arabia, *F. proliferatum* and *F. solani* can become a threat to date palm plantations [6, 7]. Saleh et al. (2017) reported that most *Fusarium* strains recovered from date palm trees showing disease symptoms were belonging to *F. proliferatum*, followed by *F. solani*. They also showed that fungal strains of *F. proliferatum* and *F. solani* had high colonization abilities on date palm leaflets.

The genetic structure and diversity of *Fusarium* strains have been assessed by genome-wide fingerprinting approaches such as amplified fragment length polymorphisms (AFLP), random amplified polymorphic DNA (RAPD) and simple sequence repeats (SSR) [8, 9]. SSR markers are tandem repeats of DNA motifs, ranging from 1 to 6 nucleotides, and highly polymorphic in eukaryotic genomes [10]. The polymorphisms of SSR mainly result from the variability in the length of repetitive units, rather than DNA sequence, and lead to multiple alleles per locus [11]. SSR markers developed for a particular species can be used to genotype other closely related species [12]. SSR markers have been successfully used for population genetics and genetic mapping studies of *Fusarium* [13]. Moreover, SSR markers have been used to differentiate between different formae speciales of *F. oxysporum* [14, 15] and to confirm the identity of *Fusarium* strains to the species level [16].

The study of variations in genetic, physiological and pathogenicity traits can help in controlling plant pathogenic fungi. However, information on the genetic variation of pathogenic *Fusarium* strains associated with Saudi date palms is very limited. The aims of the present study were to (1) use SSR markers to assess the genetic variation among *Fusarium* strains isolated from date palm trees from different geographical locations in Saudi Arabia, (2) evaluate the effect of different temperatures on mycelial growth of *Fusarium* strains *in vitro* and (3) evaluate the *in vivo* pathogenicity of pathogenic *F. proliferatum* strains on local date palm cultivars. The outcomes of the present study provide important information about the genetic diversity and pathogenicity of pathogenic *Fusarium* strains recovered from date palm-growing areas in Saudi Arabia.

## Materials and methods

### Genotyping of *Fusarium* strains using SSR markers

*Fusarium* **strains.** *Fusarium* strains, collected previously from different date palm-growing locations (Hail, Riyadh, Al-Madinah, Eastern province, Al-Qassim, Bishah and Al-Jouf) in Saudi Arabia and morphologically and molecularly characterized [7], were revived from fungal cultures preserved in 15% glycerol at -80˚C by scraping some of the ice from the top of each vial and transferred to potato dextrose agar (PDA, Difco Laboratories, Detroit, MI, USA) plates. The inoculated PDA plates were incubated at 25˚C. After 3 to 5 days, a small disc (0.3 cm in diameter) of mycelia of each strain was transferred to 50 mL Difco potato dextrose broth (PDB) in a 150-mL flask. The flasks were placed on a shaking incubator at 25˚C for 7–10 days at 100 rpm. Then, fungal mycelia were filtered through Whatman No. 2 filter papers using a vacuum machine. The dried fungal mycelia were collected from the filter papers, put in aluminum foil, wrapped and kept at -40˚C until DNA extraction [17].

**DNA extraction.** Fungal mycelia were ground under liquid nitrogen using a mortar and pestle. Then, 1/3 of 2-mL Eppendorf tubes were filled with ground mycelia. Genomic DNA

was extracted using CTAB method according to Murray and Thompson (1980) and modified by Saleh et al. (2017) [7, 18]. DNA solutions were quantified on 1% agarose gels stained with 10 μg/100 mL Acridine Orange in 0.5× TBE buffer and using Nanodrop 2000 Spectrophotometer (Thermo Fisher Scientific, USA). DNA solutions were diluted with sterilized distilled water (sdH$_2$O) to a final concentration of 20 ng/μL.

**Developing of SSR markers.** Fourteen SSR primer-pairs, ten developed for *F. verticillioides* [8] and four developed for *F. oxysporum* [15, 19], were used to genotype 59 *Fusarium* strains belonging to the following species: *F. proliferatum* (47/59), *F. solani* (9/59), *F. oxysporum* (2/59) and *F. verticillioides* (1/59). Forward primers were labeled with Applied Biosystems standard dyes, NED (yellow), VIC (green), PET (Red) and FAM (blue). The optimum annealing temperature for each SSR primer-pair was initially determined by gradient PCR using DNA of seven *Fusarium* strains: three *F. proliferatum*, two *F. solani*, one *F. oxysporum* and one *F. verticillioides*. PCR reactions contained 5 μL of 2× Go Taq Green Master Mix (Promega, WI, USA), 0.5 μL of each 10 μM primer, 0.5 μL of 20 ng/μL DNA and 3.5 μL of sdH$_2$O. The PCR conditions were as follows: an initial denaturation step at 94˚C for 4 min, followed by 35 cycles of a denaturation step at 94˚C for 40 s, an annealing step at 50–60˚C for 30 s and an extension step at 72˚C for 30 s, then a final extension step at 72˚C for 5 min in the Mastercycler® nexus gradient PCR machine (Eppendorf AG, Hamburg, Germany). After determining the optimum annealing temperature for each SSR primer-pair (Table 1), PCR reactions were set up as previously described but in a final volume of 25 μL. PCR amplification program was also as above but with the suitable annealing temperature for each SSR primer-pair. The PCR products generated from SSR primer-pairs were primarily run on 2% NuSieveTM agarose (Cambrex Bio Science, Rockland, ME, USA) gels stained with Acridine orange in 0.5× TBE buffer and visualized under UV light using the G-Box gel documentation system (Syngene, Cambridge, UK). Then, 0.5 μL of each SSR-PCR product was mixed with 9.25 μL of HiDi formamide and 0.25 μL of GeneScan® LIZ 500 size standard. The resultant mixtures were denatured at 95˚C for 5 min and chilled on ice for another 5 min. Two microliters of the previous mixture were loaded into ABI Prism® 3730 Genetic Analyzer (Applied Biosystems, Foster City, USA) and raw data were collected using genetic analyzer data collection software version 3.0 (Applied Biosystems). SSR data were analyzed with the Peak Scanner software 1.0 (Applied Biosystems).

**Analysis of SSR data.** SSR alleles were scored as "1" for the presence of a peak (allele) and "0" for its absence, assuming that peaks (alleles) with the same molecular size in different individuals were homologous. Unweighted pair grouping by mathematical averages (UPGMA) subroutine of PAUP 4.8b was used to construct dendrograms and determine the number of haplotypes among fungal strains. To quantitatively evaluate the informativeness of SSR markers, the polymorphism information content (PIC) was calculated as follows: $PICj = 1 - \Sigma_i^n P_i^2$, where i is i-th allele of the j-th SSR marker, n is the number of alleles of the j-th SSR marker and  is allele frequency [20].

## Effect of different temperatures on mycelial growth of *Fusarium* species

The effects of temperature on mycelial growth of *Fusarium* species were evaluated at different temperatures, ranging from 15 to 40˚C. Fungal strains were grown on PDA Petri plates (9 cm-diameter) by transferring 6 mm diameter disks of pure fungal cultures to the center of the plates. The plates were incubated at appropriate temperatures (15, 20, 25, 30, 35 and 40˚C). After 6 days of incubation, diameter of each colony was then measured in two perpendicular directions and an average of the two measurements was calculated after subtracting the 6 mm diameter of the colonized plug. Three PDA plates were used as replicates for each *Fusarium* strain.

**Table 1. SSR primer-pairs of *F. verticillioides* [8] and *F. oxysporum* [15, 19] used for genotyping Saudi *Fusarium* strains recovered from date palm trees.**

| Locus Name | Repeat sequence | Primer name and sequence (5′-3′) | TA[a] | N[b] | Allelic size range (bp) | PIC[c] |
|---|---|---|---|---|---|---|
| Fv-47 | (TGGTGC)n | FV-47F GCTGCTTAGTGGACCGTTTC | 59 | 5 | 195–249 | 0.355 |
| | | Fv-47R AATTGTTGGTGGAGGTGGAG | | | | |
| Fv-98 | (ATCC)n | Fv-98F AAACAAGATGCGGTCCATTC | 59 | 5 | 159–208 | 0.179 |
| | | Fv-98R GGATCGGAGGAGAATCAACA | | | | |
| Fv-114 | (GTCT)n | Fv-114F CGAATGCCTTGATCTGCTTC | ND | ND | ND | ND |
| | | Fv-114R GAGAATCCTGTTTGCGTGGT | | | | |
| Fv-120 | (TTG)n | Fv-120F GTAGCGCGGTAAGAAGATGC | 59 | 1 | 220 | 0 |
| | | Fv-120R AGTCGAAGCCCAACTGAAGA | | | | |
| Fv-140 | (CTCTG)n | Fv-140F AGGCCAGAGGGAAAGAGGTA | 57 | 6 | 209–300 | 0.301 |
| | | Fv-140R AGTTGGAAGGAAGCCCAGAG | | | | |
| Fv-269 | (TA)n | Fv-269F TGTAGAGCGTGTTCGCTTGT | ND | ND | ND | ND |
| | | Fv-269R CGTCGGAGTTGAACGATGAT | | | | |
| Fv-284 | (AAGAA)n | Fv-284F TCGGCGGGAGATTATACAAG | 59 | 4 | 213–255 | 0.145 |
| | | Fv-284R ATGGTGAACAGGAGGGACAG | | | | |
| Fv-312 | (CAGA)n | Fv-312F TTTCCGAATTCCTGGATCTG | 57 | 3 | 171–244 | 0.078 |
| | | Fv-312R GACGCAGTTTGCACAAGGTA | | | | |
| Fv-338 | (AGCAG)n | Fv-338F TAGACCAGGCAGACGAGACA | 53 | 6 | 113–272 | 0.344 |
| | | Fv-338R TGTGAGTGGGTGAGAGTGGA | | | | |
| Fv-403 | (GTGCT)n | Fv-403F GGTGTTGAGAGCGAGTGTGA | ND | ND | ND | ND |
| | | Fv-403R AGACAAGGCAAGGCAAGGTA | | | | |
| FO18 | (CAACA)n | MB18F GGTAGGAAATGACGAAGCTGAC | 57 | 4 | 277–290 | 0.113 |
| | | MB18R TGAGCACTCTAGCACTCCAAAC | | | | |
| FoAB11 | (CACAGCA)n | FoAB11F GGCCGCCCAGAAGAGGTAG | 50 | 9 | 125–269 | 0.432 |
| | | FoAB11R ATTGGAGCGGAAAAGAAACACG | | | | |
| FoDC5 | (TG)n | FoDC5F AGAAACAAGAACCCCATATCGC | 60 | 5 | 100–115 | 0.290 |
| | | FoDC5R ACTTAAACAGGAAAGGGACGGA | | | | |
| FoDD7 | (CTT)n | FoDD7F CGATTGACTACCGGGTGAACTTGT | 57 | 7 | 306–374 | 0.511 |
| | | FoDD7R AGGGCGAGGGTGAGGGTGAGA | | | | |

[a] The appropriate annealing temperature obtained from PCR gradients for each SSR primer-pair

[b] Number of alleles

[c] The polymorphism information content (PIC) was calculated according to Chesnokov and Artemyeva (2015)

ND No data

**Data analysis of temperature effect on mycelial growth of *Fusarium* species.** Data obtained from growing *Fusarium* strains at different temperatures were analyzed by one-way ANOVA using SAS software version 9.2 (SAS Institute, Cary, NC, USA). Duncan's multiple range test was performed for comparing means of different treatments of all experiments at *P < 0.05* using SAS software. To get the optimum growth temperature for each *Fusarium* species, mycelial growth was plotted against different temperatures and each curve was fitted by the polynomial regression model "y = a+bx+cx2" using Microsoft excel 2016.

## Pathogenicity of *Fusarium* strains on Saudi date palm cultivars

Three local date palm cultivars, namely Sheeshee, Khalas and Ruziz were used for pathogenicity experiments under greenhouse conditions. The seeds were surface disinfected in 1% of sodium hypochlorite for 10 min and then immersed in sdH$_2$O overnight. The soaked seeds were planted in 9×9 cm pots containing sterilized peat:soil mixture (2:3 v/v). Each pot had

only one seed. Seeded pots were irrigated and fertilized as needed under greenhouse conditions.

As the *F. proliferatum* was the dominant *Fusarium* sp. recovered from Saudi date palms, four potent pathogenic strains of *F. proliferatum* and one strain of *F. oxysporum* were further selected to infect six-month-old Sheeshee, Khalas and Ruziz seedlings under greenhouse conditions [7, 21]. PDA disks (5 mm in diameter) were cut from the edges of 5-day old cultures of *Fusarium* and transferred to 500-mL flasks containing 150 mL of PDB. Inoculated flasks were incubated and agitated on a horizontal shaker (100 rpm) at room temperature. After two weeks, the spores were collected using four layers of cheesecloth and spore filtrates were diluted with sdH$_2$O and adjusted to $1\times10^7$ conidia/mL by using a hemocytometer.

Six-month-old Sheeshee, Khalas and Ruziz seedlings were gently removed from the pots and immersed up to the crown region in $1\times10^7$ conidia/ml solutions for 10 min. Then, the inoculated seedlings were transferred to 12×15 cm pots, one seedling per pot. The control seedlings were immersed in sdH$_2$O for 10 min before transferring into the 12×15 cm pots. Four seedlings (replicas) were inoculated with each *Fusarium* strain. All pots were maintained in greenhouse until the end of the experiments. The experimental design of the greenhouse work was in a randomized complete block design (RCBD). Four weeks after seedlings' inoculation, the disease severity was estimated using the following disease rating scale: 0 = healthy (no observable disease symptoms), 1 = shrinking of one leaf, 2 = stunting and shrinking of one or two leaves, 3 = stunting and shrinking of two leaves with one leaf died, 4 = completely dead seedling (Fig 1A).

**Isolation of *Fusarium* strains from above and underground parts of infected seedlings of four date palm cultivars.** To isolate *Fusarium* strains from above and underground parts of infected seedlings, plants were gently removed from the pots and washed thoroughly under tap water. From each seedling, four plant pieces (ca 1 cm in length each) were cut, almost 1 cm apart from the crown region, from both above and underground parts. Plant pieces were sterilized for 5 min in 1% Sodium hypochlorite, dried on paper tissues and transferred to PDA plates. Four tissue pieces were arranged sequentially from 1 (close to crown region) to 4 (far from the crown region) onto the surface of PDA plates (Fig 1B). PDA plates containing plant pieces were incubated for 5 days at 25˚C. The average recovery of a fungal strain was calculated by summing plant pieces that showed the fungal growth and divided by the total number of pieces used for fungal isolation (Fig 1B).

**Data analysis of *Fusarium* pathogenicity.** The data obtained from the pathogenicity experiments of the five *Fusarium* strains on Sheeshee, Khalas and Ruziz seedlings were analyzed, using SAS software, with a two-way (*Fusarium* strains and date palm seedlings cultivars) ANOVA in RCBD with four replicates. Duncan's multiple range test was performed for comparing means of different treatments of all experiments at *P < 0.05*.

## Results

### Genotyping of *Fusarium* strains with SSR markers

A total of 14 SSR primer-pairs were used to fingerprint *Fusarium* strains recovered from date palm in Saudi Arabia. The primer-pairs Fv-114 and Fv-269 did not amplify PCR products from all *Fusarium* strains. Although Fv-403 amplified PCR products from *Fusarium* strains, bands were not clearly scorable. Collectively, 11 out of 14 SSR primer-pairs amplified a total of 57 scorable alleles from *F. proliferatum*, *F. solani*, *F. oxysporum* and *F. verticillioides* strains (Table 1). The primer-pair Fv-120 amplified only one allele from *F. verticillioides*. The number of alleles for individual SSR primer-pair ranged from 3 (Fv-312) to 9 alleles (FoAB11), with a mean number of alleles 5 per SSR marker. In addition, PIC values of SSR markers ranged from

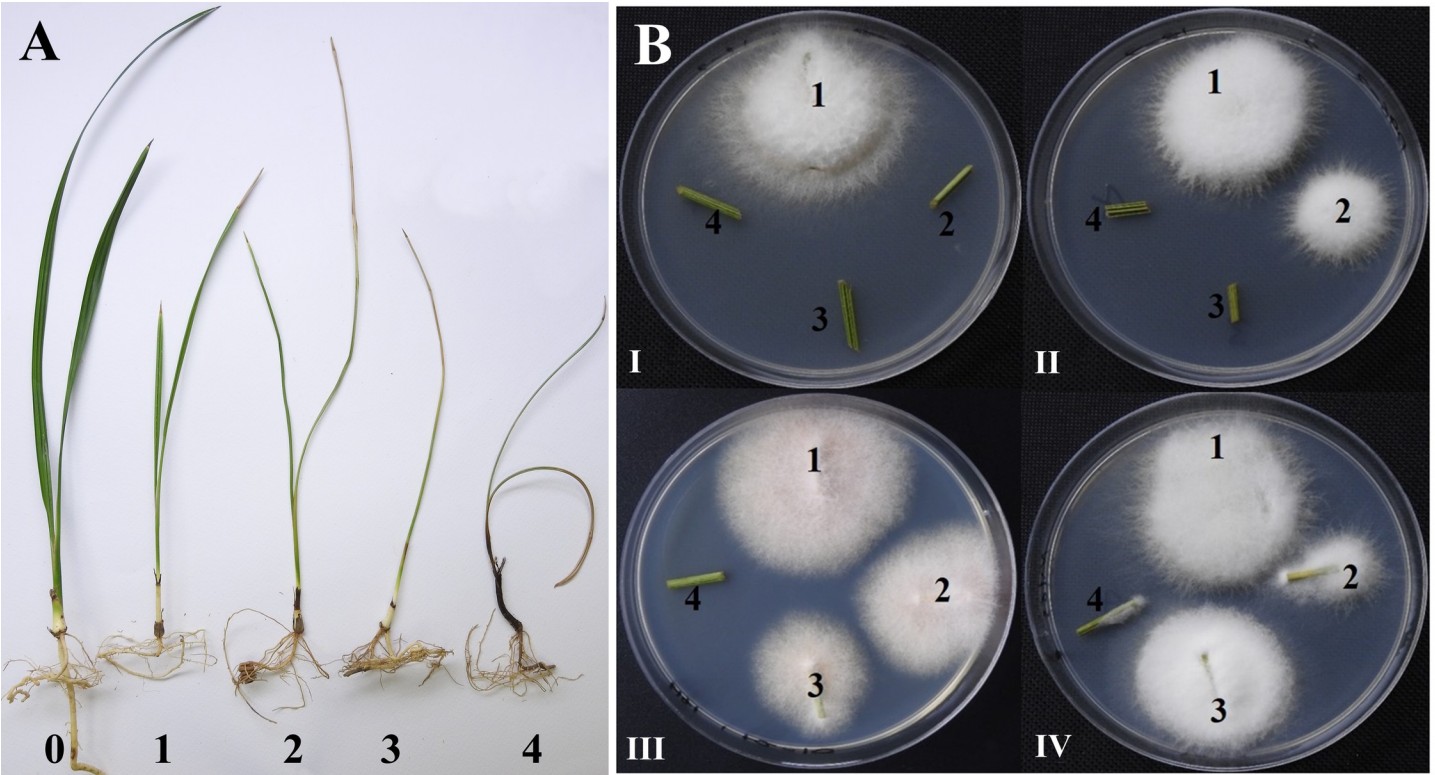

**Fig 1.** (A) Disease severity rating scale, (B) Reisolation of *Fusarium* strains from above-ground parts of inoculated date palm seedlings. (A) Disease severity rating scale from 0 to 4 according to the visible symptoms on Sheeshee seedlings inoculated with different strains of *F. proliferatum*, where 0 = healthy (no observable disease symptoms), 1 = shrinking of one leaf, 2 = stunting and shrinking of one or two leaves, 3 = stunting and shrinking of two leaves with one leaf died, 4 = completely dead seedling. (B) Reisolation of *Fusarium* strains from above-ground parts of inoculated seedlings. The average recovery of a fungal strain was calculated by summing plant pieces that showed the fungal growth and divided by the total number of pieces used for fungal isolation. Plates I, II, III, and IV show average recovery values of 1, 2, 3, and 4, respectively, for different *Fusarium* strains.

0.078 to 0.511 (Table 1). According to the number of nucleotides in SSR motifs, hexa- and penta-nucleotide repeats showed higher PIC values compared with and tetra-, tri- and di-nucleotide ones (Table 1). Among 47 *F. Proliferatum* strains, 6 out of 10 SSR markers were found to be polymorphic. The number of alleles per SSR marker varied, ranging from 1 (FoDC5, Fv-284, FO18 and Fv-312) to 5 (FoDD7), with a total of 24 alleles. The size of alleles ranged from 111 to 374 bp detected by the ABI genetic analyzer. The PIC values were low, ranging from 0.04 (Fv-338) to 0.297 (Fv-47).

SSR fingerprints of *Fusarium* strains were used to construct a phylogenetic tree using UPGMA algorithm (Fig 2). Strains of *F. proliferatum* grouped together in one clade that received

95% bootstrap value. Within *F. proliferatum* clade, 14 SSR genotypes were identified, of which nine were singleton, i.e. represented only with one strain, and five genotypes had more than one strain. Moreover, four out five genotypes contained strains isolated from more than one location. The most frequent genotype contained 21 strains isolated from root and shoot tissues collected from all the surveyed locations (Fig 2). *Fusarium verticillioides* E52 was closely related to the *F. proliferatum* clade, as both species belonging to *Fusarium fujikuroi* species complex. Most *F. solani* strains (8 out of 9) grouped in one clade that received 95% bootstrap value (Fig 2). The *F. solani* H13 strain was slightly distant from *F. solani* clade. The clade that contained the two *F. oxysporum* strains received 69% bootstrap value.

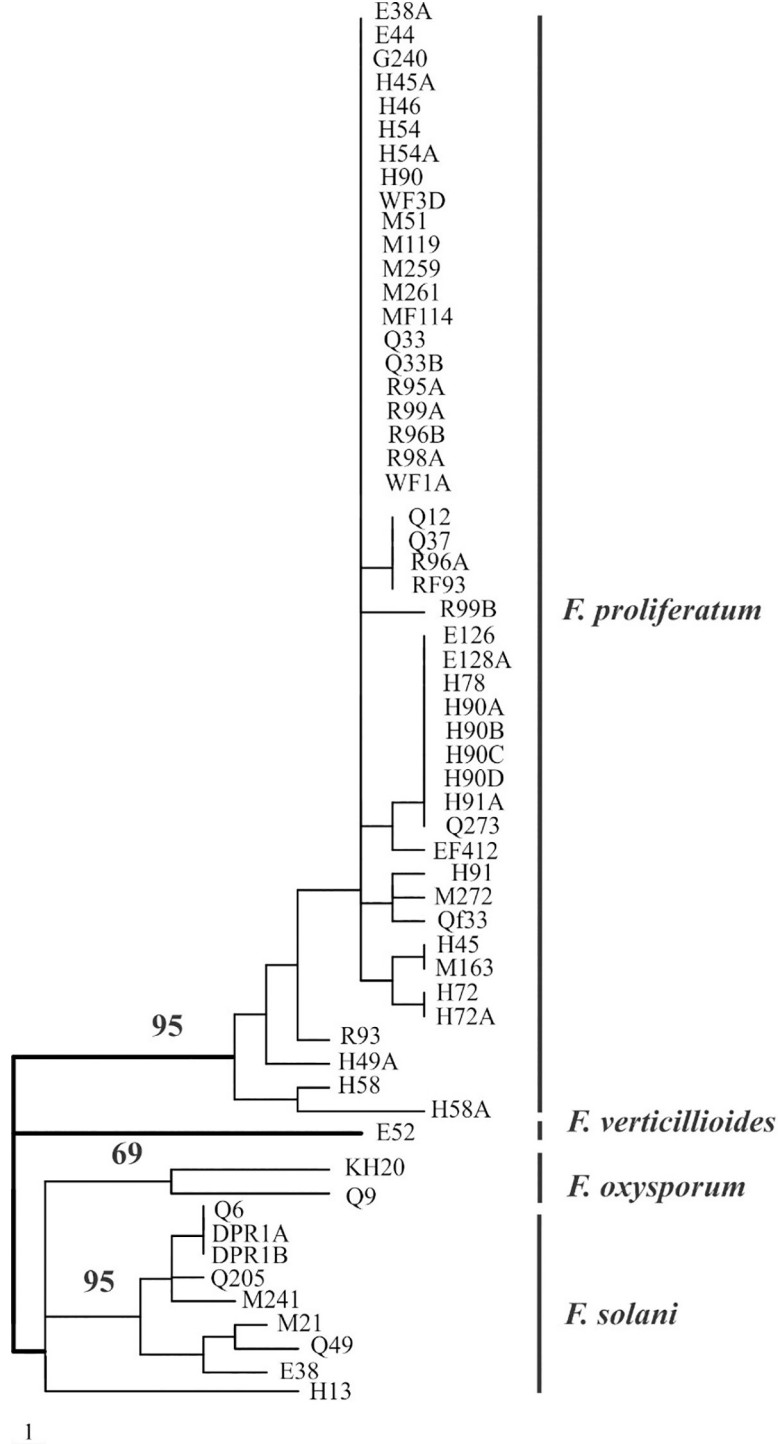

**Fig 2. An UPGMA tree showing the relationship of 59 strains of *Fusarium* isolated from date palm trees and genotyped with SSR markers.** Numbers on tree branches indicate the percentage of bootstrap values based on 1000 replicates. Bar scale at the bottom of the tree represents one SSR allele change.

## Temperature effects on mycelial growth of different *Fusarium* species

According to ANOVA, the average effects of different temperatures on fungal radial growth of *Fusarium* species were highly significant at $P < 0.0001$. In general, among different *Fusarium* species, there were differences in their radial growth. For example, at low temperatures 15 and 20˚C, *F. proliferatum* strains had the highest radial growth and *F. solani* had the lowest growth (Table 2). At moderate temperatures 25 and 30˚C, *F. verticillioides* showed the highest radial mycelial growth (Table 2). At high temperatures 35 and 40˚C, the mycelial growth of *F. solani* had the highest significant growth. *Fusarium proliferatum*, *F. verticillioides* and *F. oxysporum* didn't grow at 40˚C (Table 2). The optimum growth temperature for each *Fusarium* species was estimated using the polynomial regression model (Fig 3). *Fusarium solani* had the highest optimum temperature (27.2˚C), followed by *F. proliferatum*, *F. verticillioides* and *F. oxysporum* (25, 24.9 and 24.9˚C, respectively) (Fig 3).

Within *F. proliferatum* strains, there were significant differences in their mycelial growth on PDA ($P < 0.0001$) (Table 2). Generally, different temperatures promoted variable mycelial growth of *F. proliferatum* strains (Table 2). The highest average mycelial growth was recorded at 25˚C, followed by 20˚C and 30˚C, respectively. Both 35˚C and 15˚C temperature regimes showed lower average mycelial growth for *F. proliferatum* strains. Apparently, temperatures between 20˚C to 25˚C would be better for the growth of *F. proliferatum* strains. For *F. solani* strains, there were significant differences in mycelial growth on PDA at $P < 0.0001$ (Table 2). The highest average mycelial growth was recorded at 25˚C, followed by 30˚C. Strains of *F. solani* showed slight mycelial growth at 40˚C. It is expected that temperatures between 25˚C to 30˚C would be better for the growth of *F. solani* strains.

## Pathogenicity of *Fusarium* strains on Saudi date palm cultivars

The *F. proliferatum* H78, E128, WF3D and RF93 strains, along with *F. oxysporum* KH20 had different degrees of disease severity on the date palm seedlings of Sheeshee, Khalas and Ruziz cultivars. Disease symptoms started on the first (old) leaf that showed chlorosis and/or were shrunken, then proceeded to the second leaf. As the disease progressed, either one or two leaves died, according to strain virulence. Overall, the five *Fusarium* strains demonstrated significant differences ($P < 0.0025$) in their pathogenesis on date palm cultivars (Table 3). Two *F. proliferatum* strains RF93 and H78 had the highest disease severity values, followed by WF3D, E128 and KH20 (Table 3). Based on average effects of disease severity, there was no significant differences among date palm cultivars. However, Ruziz was slightly susceptible than Khalas and Sheeshee (Table 3).

## Isolation of *F. proliferatum* and *F. oxysporum* strains from above and underground parts of infected date palm seedlings

To investigate the fungal progress inside *Fusarium*-infected seedlings of Sheeshee, Khalas and Ruziz cultivars, direct isolation of *Fusarium* from above and underground parts were conducted. All root tissues of the three cultivars showed complete colonization by the five *Fusarium* strains mycelial. However, there were significant differences ($P < 0.0001$) in strain recovery from aboveground tissues (Table 4). The strain recovery from Ruziz was higher compared with Khalas and Sheeshee. The high recovery value could explain the slight susceptibility of Ruziz to *Fusarium* strains compared with Khalas and Sheeshee seedlings (Table 4).

## Discussion

A better understanding of genetic variation within and between phytopathogen strains would be useful in the improvement of disease management strategies of date palm trees against

**Table 2. Effect of different temperature on mycelial growth of *Fusarium* strains isolated from date palm trees in Saudi Arabia.**

| Strain* | Species | 15°C** | 20°C | 25°C | 30°C | 35°C | 40°C |
|---|---|---|---|---|---|---|---|
| E126 | *F. proliferatum* | 2.50$^{st}$ | 4.83$^{s}$ | 6.10$^{t}$ | 4.08$^{mln}$ | 2.80$^{ji}$ | 0.00$^{f}$ |
| E128A | *F. proliferatum* | 2.80$^{kml}$ | 4.33$^{ut}$ | 5.47$^{x}$ | 3.24$^{v}$ | 2.47$^{m}$ | 0.00$^{f}$ |
| E38A | *F. proliferatum* | 3.27$^{dfe}$ | 6.20$^{e}$ | 6.60$^{n}$ | 5.88$^{b}$ | 4.17$^{b}$ | 0.00$^{f}$ |
| E44 | *F. proliferatum* | 2.80$^{kml}$ | 4.93$^{r}$ | 5.73$^{wv}$ | 3.60$^{str}$ | 2.63$^{lk}$ | 0.00$^{f}$ |
| EF412 | *F. proliferatum* | 2.57$^{rsqt}$ | 4.40$^{t}$ | 5.80$^{v}$ | 3.24$^{v}$ | 2.00$^{p}$ | 0.00$^{f}$ |
| G240 | *F. proliferatum* | 3.00$^{hgi}$ | 5.27$^{op}$ | 7.20$^{g}$ | 4.72$^{kij}$ | 3.63$^{edf}$ | 0.00$^{f}$ |
| H45 | *F. proliferatum* | 3.27$^{dfe}$ | 6.03$^{f}$ | 8.40$^{a}$ | 6.28$^{a}$ | 4.70$^{a}$ | 0.00$^{f}$ |
| H45A | *F. proliferatum* | 2.93$^{hji}$ | 5.97$^{gf}$ | 6.70$^{m}$ | 4.24$^{l}$ | 2.63$^{lk}$ | 0.00$^{f}$ |
| H46 | *F. proliferatum* | 3.37$^{dc}$ | 6.33$^{d}$ | 7.87$^{d}$ | 5.92$^{b}$ | 3.67$^{ed}$ | 0.00$^{f}$ |
| H49A | *F. proliferatum* | 2.83$^{kjl}$ | 5.67$^{ij}$ | 8.40$^{a}$ | 6.00$^{b}$ | 3.83$^{c}$ | 0.00$^{f}$ |
| H54 | *F. proliferatum* | 2.57$^{rsqt}$ | 5.23$^{op}$ | 6.33$^{r}$ | 3.48$^{stu}$ | 2.73$^{jk}$ | 0.00$^{f}$ |
| H58 | *F. proliferatum* | 3.03$^{hg}$ | 5.33$^{no}$ | 7.20$^{g}$ | 4.76$^{hkij}$ | 3.00$^{h}$ | 0.00$^{f}$ |
| H58A | *F. proliferatum* | 3.60$^{a}$ | 5.53$^{k}$ | 7.23$^{g}$ | 4.80$^{hij}$ | 2.60$^{l}$ | 0.00$^{f}$ |
| H72 | *F. proliferatum* | 2.47$^{t}$ | 5.23$^{op}$ | 6.50$^{op}$ | 4.20$^{ml}$ | 2.87$^{i}$ | 0.00$^{f}$ |
| H78 | *F. proliferatum* | 2.77$^{nml}$ | 3.83$^{w}$ | 4.60$^{y}$ | 3.36$^{vu}$ | 2.10$^{po}$ | 0.00$^{f}$ |
| H90 | *F. proliferatum* | 3.23$^{fe}$ | 5.30$^{nop}$ | 7.27$^{g}$ | 4.04$^{mon}$ | 3.33$^{g}$ | 0.00$^{f}$ |
| H90A | *F. proliferatum* | 2.60$^{rsqp}$ | 6.00$^{f}$ | 6.97$^{ih}$ | 3.92$^{pon}$ | 2.63$^{lk}$ | 0.00$^{f}$ |
| H90B | *F. proliferatum* | 2.53$^{rst}$ | 5.63$^{j}$ | 6.90$^{ij}$ | 4.12$^{ml}$ | 3.00$^{h}$ | 0.00$^{f}$ |
| H90C | *F. proliferatum* | 2.93$^{hji}$ | 5.90$^{g}$ | 6.73$^{ml}$ | 3.88$^{po}$ | 2.87$^{i}$ | 0.00$^{f}$ |
| H90D | *F. proliferatum* | 2.57$^{rsqt}$ | 6.00$^{f}$ | 7.37$^{f}$ | 3.80$^{pq}$ | 2.40$^{m}$ | 0.00$^{f}$ |
| H91 | *F. proliferatum* | 2.63$^{roqp}$ | 5.07$^{q}$ | 6.23$^{s}$ | 3.44$^{tu}$ | 2.17$^{o}$ | 0.00$^{f}$ |
| H91A | *F. proliferatum* | 2.67$^{noqp}$ | 5.53$^{k}$ | 6.87$^{kj}$ | 3.76$^{pqr}$ | 3.00$^{h}$ | 0.00$^{f}$ |
| M119 | *F. proliferatum* | 2.73$^{noml}$ | 5.20$^{p}$ | 5.67$^{w}$ | 3.64$^{sqr}$ | 2.00$^{p}$ | 0.00$^{f}$ |
| M163 | *F. proliferatum* | 3.20$^{f}$ | 6.77$^{b}$ | 8.40$^{a}$ | 6.36$^{a}$ | 3.33$^{g}$ | 0.00$^{f}$ |
| M259 | *F. proliferatum* | 2.70$^{omp}$ | 6.20$^{e}$ | 7.40$^{f}$ | 5.16$^{fe}$ | 2.60$^{l}$ | 0.00$^{f}$ |
| M261 | *F. proliferatum* | 3.03$^{hg}$ | 5.73$^{ih}$ | 7.00$^{h}$ | 5.28$^{de}$ | 3.73$^{d}$ | 0.00$^{f}$ |
| M272 | *F. proliferatum* | 3.33$^{dce}$ | 6.57$^{c}$ | 7.50$^{e}$ | 5.28$^{de}$ | 3.53$^{f}$ | 0.00$^{f}$ |
| M51 | *F. proliferatum* | 3.5$^{0b}$ | 6.30$^{d}$ | 8.03$^{c}$ | 5.96$^{b}$ | 3.57$^{ef}$ | 0.00$^{f}$ |
| MF114 | *F. proliferatum* | 3.00$^{hgi}$ | 5.43$^{lm}$ | 6.80$^{kl}$ | 4.20$^{ml}$ | 3.33$^{g}$ | 0.00$^{f}$ |
| Q12 | *F. proliferatum* | 2.33$^{u}$ | 5.63$^{j}$ | 6.97$^{ih}$ | 4.60$^{k}$ | 3.30$^{g}$ | 0.00$^{f}$ |
| Q273 | *F. proliferatum* | 2.90$^{kji}$ | 6.00$^{f}$ | 6.73$^{ml}$ | 5.40$^{dc}$ | 3.90$^{c}$ | 0.00$^{f}$ |
| Q33 | *F. proliferatum* | 2.83$^{kjl}$ | 5.43$^{lm}$ | 6.80$^{kl}$ | 4.60$^{k}$ | 3.40$^{g}$ | 0.00$^{f}$ |
| Q33B | *F. proliferatum* | 3.37$^{dc}$ | 6.93$^{a}$ | 7.90$^{d}$ | 4.80$^{hij}$ | 2.60$^{l}$ | 0.00$^{f}$ |
| QF33 | *F. proliferatum* | 3.53$^{ba}$ | 6.97$^{a}$ | 8.13$^{b}$ | 5.56$^{c}$ | 3.60$^{ef}$ | 0.00$^{f}$ |
| R93 | *F. proliferatum* | 2.50$^{st}$ | 5.80$^{h}$ | 7.40$^{f}$ | 4.64$^{kj}$ | 3.33$^{g}$ | 0.00$^{f}$ |
| R95A | *F. proliferatum* | 3.40$^{c}$ | 5.47$^{lk}$ | 6.60$^{n}$ | 3.68$^{qr}$ | 2.83$^{ji}$ | 0.00$^{f}$ |
| R96A | *F. proliferatum* | 3.23$^{fe}$ | 5.23$^{op}$ | 5.97$^{u}$ | 3.48$^{stu}$ | 2.60$^{l}$ | 0.00$^{f}$ |
| R96B | *F. proliferatum* | 2.90$^{kji}$ | 5.30$^{nop}$ | 6.37$^{qr}$ | 3.36$^{vu}$ | 2.57$^{l}$ | 0.00$^{f}$ |
| R98A | *F. proliferatum* | 2.77$^{nml}$ | 5.00$^{rq}$ | 6.43$^{qp}$ | 4.92$^{hg}$ | 2.83$^{ji}$ | 0.00$^{f}$ |
| R99A | *F. proliferatum* | 2.63$^{roqp}$ | 4.27$^{u}$ | 4.23$^{z}$ | 3.24$^{v}$ | 2.30$^{n}$ | 0.00$^{f}$ |
| R99B | *F. proliferatum* | 2.60$^{rsqp}$ | 3.97$^{v}$ | 3.83$^{a}$ | 2.68$^{w}$ | 2.03$^{p}$ | 0.00$^{f}$ |
| RF93 | *F. proliferatum* | 3.03$^{hg}$ | 5.37$^{nm}$ | 6.57$^{on}$ | 4.08$^{mln}$ | 2.87$^{i}$ | 0.00$^{f}$ |
| WF1A | *F. proliferatum* | 3.07$^{g}$ | 5.23$^{op}$ | 6.50$^{op}$ | 4.84$^{hi}$ | 3.63$^{edf}$ | 0.00$^{f}$ |
| WF3D | *F. proliferatum* | 2.47$^{t}$ | 5.50$^{lk}$ | 6.40$^{qr}$ | 5.04$^{fg}$ | 3.33$^{g}$ | 0.00$^{f}$ |
| Average effect of *F. proliferatum*** | | 2.91$^{E}$ | 5.52$^{B}$ | 6.73$^{A}$ | 4.45$^{C}$ | 3.01$^{D}$ | 0.00$^{F}$ |
| KH20 | *F. oxysporum* | 2.33$^{b}$ | 5.27$^{a}$ | 6.73$^{a}$ | 3.92$^{a}$ | 1.83$^{a}$ | 0.00$^{f}$ |
| Q9 | *F. oxysporum* | 2.56$^{a}$ | 5.00$^{b}$ | 4.80$^{b}$ | 3.56$^{a}$ | 1.93$^{a}$ | 0.00$^{f}$ |

*(Continued)*

**Table 2.** (Continued)

| Strain* | Species | 15°C** | 20°C | 25°C | 30°C | 35°C | 40°C |
|---|---|---|---|---|---|---|---|
| Average effect of *F. oxysporum* | | 2.45$^D$ | 5.13$^B$ | 5.77$^A$ | 3.74$^C$ | 1.88$^E$ | 0.00$^F$ |
| E52 | *F. verticillioides* | 2.43$^E$ | 5.57$^C$ | 7.23$^A$ | 6.04$^B$ | 3.43$^D$ | 0.00$^F$ |
| DPR1A | *F. solani* | 1.63$^b$ | 4.30$^b$ | 6.27$^b$ | 5.52$^d$ | 5.23$^b$ | 0.27$^d$ |
| DPR1B | *F. solani* | 1.50$^c$ | 4.23$^b$ | 6.20$^b$ | 5.40$^e$ | 5.50$^a$ | 0.33$^{dc}$ |
| E38 | *F. solani* | 0.80$^e$ | 3.33$^e$ | 5.97$^d$ | 6.00$^b$ | 5.57$^a$ | 1.40$^a$ |
| H13 | *F. solani* | 3.33$^a$ | 6.73$^a$ | 7.37$^a$ | 5.64$^c$ | 4.63$^e$ | 0.07$^e$ |
| M21 | *F. solani* | 1.37$^d$ | 4.03$^c$ | 6.07$^c$ | 6.24$^a$ | 5.03$^c$ | 0.63$^b$ |
| M241 | *F. solani* | 1.50$^c$ | 4.20$^b$ | 5.80$^e$ | 5.16$^f$ | 4.83$^d$ | 0.07$^e$ |
| Q205 | *F. solani* | 1.50$^c$ | 3.90$^d$ | 5.10$^g$ | 4.96$^h$ | 5.00$^c$ | 0.40$^c$ |
| Q49 | *F. solani* | 0.73$^f$ | 3.03$^f$ | 5.00$^h$ | 5.04$^{gh}$ | 5.23$^b$ | 0.40$^c$ |
| Q6 | *F. solani* | 1.50$^c$ | 3.33$^e$ | 5.47$^f$ | 5.12$^{gf}$ | 5.00$^c$ | 0.43$^c$ |
| Average effect of *F. solani* | | 1.54$^D$ | 4.12$^C$ | 5.91$^A$ | 5.45$^B$ | 5.11$^B$ | 0.44$^E$ |

*The first letter of strain ID refers to the location from which it was isolated as follows: E (Eastern province), B (Bishah), G (Al-Jouf), H (Hail), M (Al-Madinah), Q (Al-Qassim), R (Riyadh) and W (Wadi Ad-Dawasir).

**For each *Fusarium* species, values within a column and followed by the same small letters are not statistically significant from each other.

***For the average effect of each *Fusarium* species, values within a row and followed by the same capital letters are not statistically significant from each other.

pathogenic *Fusarium* spp. Among DNA-based methods used to assess genetic variations of organisms, SSR loci become the markers of choice for studying genetic diversity of organisms. Many studies have also evaluated the transferability of SSR across species and genera of different kingdoms including fungi and plants [22]. In this study, SSR markers previously developed for *F. verticillioides* and *F. oxysporum* have been used to determine genetic diversity of *Fusarium* strains recovered from date palm trees in Saudi Arabia. The four SSR primer-pairs of *F. oxysporum* amplified fingerprints from Saudi *Fusarium* strains. However, eight out of the ten *F. verticillioides* SSR primer-pairs amplified PCR amplicons from *Fusarium* strains. Although,

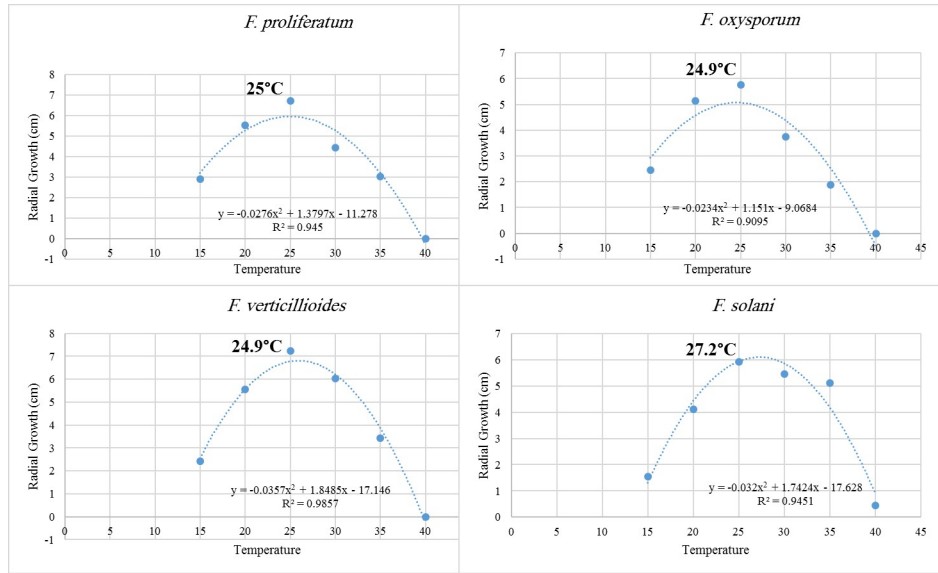

**Fig 3. Estimation of optimum mycelial growth of four *Fusarium* species grown at different temperatures using the polynomial regression model.**

**Table 3. Disease severity of five *Fusarium* strains on date palm seedlings of three local cultivars.**

| Strain* | Seedlings** | | |
|---|---|---|---|
| | **Khalas** | **Ruziz** | **Sheeshee** |
| E128A | 0.75[ab] | 2.50[a] | 2.50[a] |
| H78 | 1.75[ab] | 3.25[a] | 1.25[ab] |
| RF93 | 3.00[a] | 1.25[ab] | 2.25[a] |
| WF3D | 2.50[a] | 2.25[a] | 1.25[ab] |
| KH20 | 1.50[ab] | 2.25[a] | 0.75[ab] |
| Control | 0.00[b] | 0.00[b] | 0.00[b] |
| Average effect | 1.58[A] | 1.92[A] | 1.33[A] |

*\**Fusarium* strains E128, H78, RF93 and WF3D belong to *F. proliferatum*, whereas KH20 strain belongs to *F. oxysporum*.

**\*\*Disease severity on date palm seedlings was estimated 4 weeks after inoculation using a 0–4 scale (Fig 1A). Values within a column followed by the same small letters are not statistically significant from each other. For the main average effect, values within a row followed by the capital letters are not statistically significant from each other.

primer-pair Fv-120 amplified a PCR amplicon from Saudi *F. verticillioides* strain, it did not generate any PCR amplicons from other *Fusarium* spp. The obtained data are in agreement with previous studies on different *Fusarium* species [13, 23]. For example, Saharan and Naef (2008) used 10 SSR markers developed for *F. graminearum* to detect genetic diversity of *F. oxysporum* and *F. verticillioides*, along with *F. graminearum* isolates, recovered from naturally infected wheat in India and found that not all the *F. graminearum* SSR markers amplified PCR products from *F. oxysporum* and *F. verticillioides* isolates. Moncrief et al. (2016) developed 17 SSR markers to differentiate pathogenic strains of *F. proliferatum* collected from onion tissues displaying salmon blotch symptoms from other pathogenic strains representing different hosts and countries, as well as from other *Fusarium* isolates belonging to *F. verticillioides*, *F. thapsinum*, *F. subglutinans*, *F. andiyazi*, *F. globosum*, *F. fujikoroi* and *F. oxysporum*.

In general, SSR markers are less abundant in fungal genomes compared with other organisms' genomes [24]. Fungal genomes also have unique distribution and occurrence of different

**Table 4. Re-isolation of five *Fusarium* strains from aboveground parts of inoculated date palm seedlings derived from seeds of three date palm cultivars.**

| Strain* | Seedlings** | | |
|---|---|---|---|
| | **Khalas** | **Ruziz** | **Sheeshee** |
| E128A | 2.75[ab] | 3.75[a] | 3.25[ab] |
| H78 | 3.75[a] | 4.00[a] | 3.25[ab] |
| RF93 | 2.00[b] | 3.25[a] | 3.50[a] |
| WF3D | 3.50[a] | 3.75[a] | 3.00[ab] |
| KH20 | 3.00[ab] | 3.50[a] | 2.00[b] |
| Control | 0.00[c] | 0.00[b] | 0.00[c] |
| Average effect | 2.50[B] | 3.04[A] | 2.50[B] |

*\**Fusarium* strains E128, H78, RF93 and WF3D belong to *F. proliferatum*, whereas KH20 strain belongs to *F. oxysporum*.

**\*\*Average recovery of a fungal strain was calculated by summing plant pieces that showed the fungus divided by the total number of pieces (Fig 1B). Values within a column followed by the same small letters are not statistically significant from each other. Main average effect values for the seedlings and strains followed by the same capital letters are not statistically significant from each other.

SSR motifs [24]. For example, di- and tetra-nucleotide SSR motifs are less frequent in fungi [8, 24]. In *Fusarium*, di- and tri-nucleotides SSR are mainly used to assess genetic diversity in its populations [13, 16]. However, Leyva-Madrigal et al. (2014) found that penta- and hexanucleotides are the most abundant microsatellites in the *F. verticillioides* genome. In this study, penta- and hexa-nucleotide repeats showed higher PIC values compared with and tetra-, tri- and di-nucleotide ones. The number of alleles for individual SSR primer-pair ranged from 3 (Fv-312) to 9 alleles (FoAB11), with a mean of 5 alleles per locus. The number of detected alleles in this study were lower than the number of alleles obtained previously for *F. verticillioides* and *F. oxysporum* strains [8, 15, 19]. The explanation of getting lower number of alleles could be due to the power of SSR transferability.

Based on the PIC values, Botstein et al. (1980) categorized molecular markers into three groups: (1) highly informative markers with PIC values > 0.50, (2) reasonably informative markers with PIC values between 0.50 and 0.25 and (3) slightly informative markers with PIC values < 0.25 [25]. Five of the SSR markers used in this study were reasonably informative, among *Fusarium* species, with PIC values between 0.43 and 0.29. FoDD7 marker was highly informative among *Fusarium* species. However, within *F. proliferatum* strains, two SSR markers were reasonably informative (Fv-47 and FoDD7) and two were slightly informative (Fv-140 and FoAB11). Other than the transferability of the SSR markers, the low PIC values obtained in this study could be due to the sample size of *Fusarium* species. Kalinowski (2004) reported that there is a proportional relationship between the number of alleles and population sample size [26].

When the SSR fingerprints used to construct a phylogenetic tree using UPGMA algorithm, *Fusarium* strains of each species clustered in separate clades supported with bootstrap values ≥ 69%. Although *F. proliferatum* and *F. verticillioides* are closely related, SSR markers distinguished strains of the two species. SSR markers have been successfully used to distinguish many closely related species of *Fusarium*, including *F. graminearum* and *F. pseudograminearum* [27]; *F. asiaticum* and *F. graminearum* [28] and *F. culmorum* and *F. crookwellense* [16]. Moreover, many formae speciales of *F. oxysporum* were distinguishable by SSR markers [14, 15].

Among various environmental factors that impact disease development in the host plants, the temperature has significant influences on the pathogen-host interaction. Many studies have shown that the temperature can affect the growth and multiplication of pathogens as well as the development of diseases they cause [29, 30]. In this study, different temperatures affected the radial growth of *Fusarium* strains (*P < 0.0001*) on PDA plates, with the best growth obtained at 25˚C. Indeed, many studies showed that 25˚C was most suitable temperature for *Fusarium* mycelial growth [31, 32]. At both low (15–20˚C) and high (30–35˚C) temperature regimes, *Fusarium* strains showed low average mycelial growth. At the 40˚C, no mycelial growth was observed for *F. proliferatum*, *F. verticillioides* and *F. oxysporum* strains. Similarly, Mogensen et al. (2009) reported that strains of five *Fusarium* species, including *F. proliferatum*, *F. verticillioides* and *F. oxysporum*, were not able to grow at temperatures ≥ 40˚C [33]. The only *Fusarium* strains that showed mycelial growth at 40˚C belonged to *F. solani*. It is well known that strains of *F. solani* are the most common plant pathogens in tropical and temperate regions [17, 34]. Saremi and Burgess (2000) studied the population dynamics of five *Fusarium* species representing different climatic conditions at three levels of temperatures and found that *F. solani* populations had the highest mycelial growth at high temperatures. In contrast, Ramteke and Kamble (2011) showed that *F. solani* strains causing rhizome rot of ginger were not able to grow at 40˚C.

The results from this study indicate that strains of different *Fusarium* species have different optimum growth temperatures. For example, *F. solani* had the highest optimum growth

temperature (27.1˚C), while the optimum growth temperatures for *F. verticillioides*, *F. proliferatum* and *F. oxysporum* were 25.5, 24.6 and 24.2˚C, respectively. Many *Fusarium* species favor particular environments with specific temperature ranges [17, 35]. Samapundo et al. (2005) reported that *F. verticillioides* has optimal temperature higher than *F. proliferatum* [36]. The present study showed the same trend where *F. verticillioides* showed faster mycelial growth at temperatures 25 and 30˚C compared with other tested *Fusarium* strains. Moreover, Reid et al. (1999) demonstrated that *F. verticillioides* can grow well above 28˚C [37]. The optimum temperature for mycelial growth of Saudi *F. proliferatum* recovered from date palms was 24.6˚C. Interestingly, the optimum growth temperature for *F. proliferatum* strains, isolated from garlic bulb rot in Egypt and from wheat grains in Argentina, was 25˚C [31, 38]. However, Samapundo et al. (2005) and Marín et al. (1999) showed that the optimum growth temperature for *F. proliferatum* strains isolated from maize grains was 30˚C [36, 39]. High temperatures are associated with higher levels of *Fusarium* wilt on several crops. For instance, Landa et al. (2006) noted that early seed sowing that coincided with low temperatures suppressed Fusarium wilt of chickpea, whereas late sowing that coincided with warm temperatures increased disease severity. Also, *F. solani* f. sp. *pisi*, the causal agent of chickpea root rot, was very severe on plants grown at 30˚C, whereas the plants grew at lower temperatures (10, 15, 20 and 25˚C) showed less symptoms [40]. Likewise, in a field study, *F. solani* colonized perennial ryegrass roots and produced more propagules under high temperature regimes [35].

Recently, *F. proliferatum* has become the most important pathogenic *Fusarium* species on date palms not only in Saudi Arabia but also worldwide [5, 6, 41, 42]. In the present work, pathogenicity of four potent pathogenic strains of *F. proliferatum* and one *F. oxysporum* strain were conducted on date palm seedlings generated from seeds belonging to three local cultivars (Khalas, Ruziz and Sheeshee). These strains were previously isolated from diseased trees and they also showed high colonization ability on detached leaf experiments [7]. Moreover, Sharafaddin et al. (2019) showed that these four strains of *F. proliferatum* produced different cell wall degrading enzymes enabling them to colonize leaflet cuttings of date palm tissues. Indeed, we found that these strains caused yellowing, shrinking and drying of leaves, and eventually the death of inoculated seedlings. The symptoms resembled those produced by *F. proliferatum* strains on seedlings of *P. canariensis* [43]. In some cases, different symptoms can develop on *F. proliferatum*-inoculated date palm seedlings, such as leaf blight [6]. The symptoms produced by *F. proliferatum* on majesty palm seedlings included browning areas on unopened leaves along with reddish leaf spots [44]. *Fusarium proliferatum* can cause other disease symptoms on date palm trees, e.g. bunch fading, inflorescence rot and fruit spots [41, 42]. The ability of *F. proliferatum* strains to produce a wide spectrum of disease symptoms is referred to its ability to attack a wide range of host plants [17]. Moreover, *F. proliferatum* strains are able to secrete a wide spectrum of mycotoxins that play essential roles in their pathogenicity and ecology [6]. Testing four potent pathogenic strains of *F. proliferatum* and one strain of *F. oxysporum* on local cultivars confirmed their virulence ability. The tested fungal strains represented different locations and were isolated from different date palm cultivars [7].

## Conclusions

In conclusion, the SSR markers developed for *F. verticillioides* and *F. oxysporum* were very useful in assessing the genetic diversity of *Fusarium* strains collected from Saudi date palm trees. Multi-individual SSR genotypes of *F. proliferatum* contained strains isolated from more than one region, suggesting that infected date palm tissues and/or *Fusarium*-contaminated tools are moving among different regions in Saudi Arabia. The results of the temperature study showed that the *Fusarium* strains can grow under a wide range of temperatures. This ability enables

them to survive in the extreme temperatures over different seasons. This study also provides an evidence that strains of *F. proliferatum* can consistently cause disease to date palm seedlings under controlled conditions.

## Author Contributions

**Conceptualization:** Amgad A. Saleh.

**Data curation:** Anwar H. Sharafaddin.

**Formal analysis:** Anwar H. Sharafaddin, Mahmoud H. El_Komy, Yasser E. Ibrahim, Younis K. Hamad.

**Funding acquisition:** Amgad A. Saleh.

**Methodology:** Anwar H. Sharafaddin.

**Project administration:** Amgad A. Saleh.

**Supervision:** Amgad A. Saleh.

**Writing – original draft:** Amgad A. Saleh, Anwar H. Sharafaddin, Mahmoud H. El_Komy, Yasser E. Ibrahim, Younis K. Hamad.

**Writing – review & editing:** Amgad A. Saleh, Anwar H. Sharafaddin, Mahmoud H. El_Komy, Yasser E. Ibrahim, Younis K. Hamad.

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
