## [Decision Letter · Decision Letter 0]

18 May 2021

PONE-D-21-10726

Molecular and physiological characterization of Fusarium strains associated with different diseases in date palm

PLOS ONE

Dear Dr. Saleh

Thank you for submitting your manuscript to PLOS ONE. After careful consideration, we feel that it has merit but does not fully meet PLOS ONE’s publication criteria as it currently stands. Therefore, we invite you to submit a revised version of the manuscript that addresses the points raised during the review process.

We look forward to receiving your revised manuscript.

Kind regards,

Himanshu Sharma

Academic Editor

PLOS ONE

Additional Editor Comments (if provided):

The manuscript entitled Molecular and physiological characterization of Fusarium strains associated with different diseases in date palm by Amgad A Saleh has been reviewed by the reviewers and also me, as there are many drawbacks as there is no novelty regarding SSRs markers as they have used markers from public domain, samples are also very less. So based on reviewers recommendations, manuscript can be considered for publication after addressing the reviewers comments.

Journal Requirements:

Reviewers' comments:

Reviewer's Responses to Questions

**Comments to the Author**

1. Is the manuscript technically sound, and do the data support the conclusions?

Reviewer #1: Yes

Reviewer #2: Partly

Reviewer #3: Yes

2. Has the statistical analysis been performed appropriately and rigorously? 

Reviewer #1: I Don't Know

Reviewer #2: Yes

Reviewer #3: Yes

3. Have the authors made all data underlying the findings in their manuscript fully available?

Reviewer #1: Yes

Reviewer #2: No

Reviewer #3: Yes

4. Is the manuscript presented in an intelligible fashion and written in standard English?

Reviewer #1: Yes

Reviewer #2: Yes

Reviewer #3: Yes

5. Review Comments to the Author

Reviewer #1: The article "Molecular and physiological characterization of Fusarium strains associated with different diseases in date palm" describes the identification and characterization of fusarium strains in Saudi Arabia. The authors have analyzed the effect of temperature on mycelial growth moreover also evaluated the pathogenicity of these strains on

different palm tree cultivars. These fungal strains have also been validated using few markers. In my opinion the findings of the manuscript are good but not sufficient for publication in PLos One.

Reviewer #2: The article presented the work on Fusarium species attacking date palm plants of Saudi Arabia. The concept was good, however manuscript suffers with many limitations which need to be work out. Authors used previously developed SSR markers for genetic variability of 59 Fusarium samples, tested effect of temperature on growth of fusarium and tested pathogenicity of fusarium. The objectives were clear and achieved well. Methodology require refinement at many places. there is some repetition discussion part from introduction it needs to be removed and conclusion require explanation and inferences in scientific facts rather than simple generalizations.

Reviewer #3: The MS PONE-D-21-10726 Molecular and physiological characterization of Fusarium strains associated with different diseases in date palm can be accepted for publication. However, at this stage MS need major revision. Discussion part is unnecessarily stretched. Therefore, better it should be concise. In methodology the

Abstract:

Line 30-32; authors can remove these lines from here or rewrite it for their continuity with earlier work.

Material and methods

Page no.4 Line 79-81 Cultures were preserved not maintained in -80. Rewrite these lines.

Page no. 5 Line 102 to 106; correct 0.5 μL 20 to 0.5 μL of 20…. Similarly at other places.

Page no 6 Table 1 remove why 121 SSR primer-pairs of used for genotyping will be sufficient. Author can cite the references in text.

Page no. 10 Authors should describe Pathogenicity o 154 f Fusarium strains on Saudi date palm cultivars under a single heading. No need to give subpara..

Plant material, Fungal inoculum and Seedlings inoculation should be merged

Rewrite discussion and it should be concised

6. PLOS authors have the option to publish the peer review history of their article (what does this mean?). If published, this will include your full peer review and any attached files.

Reviewer #1: **Yes: **Rajdeep Jaswal

Reviewer #2: No

Reviewer #3: No

---

## [Author Response · Author response to Decision Letter 0]

22 May 2021

Reviewer #2: The article presented the work on Fusarium species attacking date palm plants of Saudi Arabia. The concept was good, however manuscript suffers with many limitations which need to be work out. Authors used previously developed SSR markers for genetic variability of 59 Fusarium samples, tested effect of temperature on growth of fusarium and tested pathogenicity of fusarium. The objectives were clear and achieved well. Methodology require refinement at many places. There is some repetition discussion part from introduction it needs to be removed and conclusion require explanation and inferences in scientific facts rather than simple generalizations.

Response

We went through the Methodology and fixed it. Also, the repeated text was removed from the Discussion section. We tried our best to be realistic in the Conclusion.

Reviewer #3: The MS PONE-D-21-10726 Molecular and physiological characterization of Fusarium strains associated with different diseases in date palm can be accepted for publication. However, at this stage MS need major revision. Discussion part is unnecessarily stretched. Therefore, better it should be concise.

Abstract:

Line 30-32; authors can remove these lines from here or rewrite it for their continuity with earlier work.

Response

The statement was removed and added after the SSR results, please see the Abstract lines 27-29

Material and methods:

Page no.4 Line 79-81 Cultures were preserved not maintained in -80. Rewrite these lines.

Response

Fulfilled

Page no. 5 Line 102 to 106; correct 0.5 μL 20 to 0.5 μL of 20…. Similarly at other places.

Response

Fulfilled

Page no 6 Table 1 remove why 121 SSR primer-pairs of used for genotyping will be sufficient. Author can cite the references in text.

Response

The SSR references have been already cited in the text, please see Lines 98-99

Page no. 10 Authors should describe Pathogenicity o 154 f Fusarium strains on Saudi date palm cultivars under a single heading. No need to give subpara.

Plant material, Fungal inoculum and Seedlings inoculation should be merged

Response

Fulfilled

Rewrite discussion and it should be concise

Response

The Discussion section was revised

---

## [Decision Letter · Decision Letter 1]

22 Jun 2021

Molecular and physiological characterization of Fusarium strains associated with different diseases in date palm

PONE-D-21-10726R1

Dear Dr. Saleh,

We’re pleased to inform you that your manuscript has been judged scientifically suitable for publication and will be formally accepted for publication once it meets all outstanding technical requirements.

Kind regards,

Himanshu Sharma

Academic Editor

PLOS ONE

Additional Editor Comments (optional):

The manuscript entitled Molecular and physiological characterization of Fusarium strains associated with different diseases in date palm by Dr. Amgad A Saleh has been extensively reviewed by the reviewers and the authors had positively answered all the questions. So this time I am positive for the acceptance of the manuscript, as there are always chances of improvement So authors check again the mistakes and rectify them in the proof reads.

Reviewers' comments:

Reviewer's Responses to Questions

**Comments to the Author**

1. If the authors have adequately addressed your comments raised in a previous round of review and you feel that this manuscript is now acceptable for publication, you may indicate that here to bypass the “Comments to the Author” section, enter your conflict of interest statement in the “Confidential to Editor” section, and submit your "Accept" recommendation.

Reviewer #2: All comments have been addressed

Reviewer #3: All comments have been addressed

2. Is the manuscript technically sound, and do the data support the conclusions?

Reviewer #2: Partly

Reviewer #3: Yes

3. Has the statistical analysis been performed appropriately and rigorously? 

Reviewer #2: Yes

Reviewer #3: Yes

4. Have the authors made all data underlying the findings in their manuscript fully available?

Reviewer #2: Yes

Reviewer #3: Yes

5. Is the manuscript presented in an intelligible fashion and written in standard English?

Reviewer #2: Yes

Reviewer #3: Yes

6. Review Comments to the Author

Reviewer #2: The authors have addressed all the raised issues and manuscript can be accepted now however, authors are advised to go through the final version to remove any type of lingual and technical errors.

Reviewer #3: All the comments raised in earlier revision have been addressed by the authors pointwise. Therefore, MS can be accepted for publication in my opinion.

7. PLOS authors have the option to publish the peer review history of their article (what does this mean?). If published, this will include your full peer review and any attached files.

Reviewer #2: **Yes: **Dr. Vikas Sharma

Reviewer #3: **Yes: **vivek sharma

---

## [Editor Report · Acceptance letter]

14 Jul 2021

PONE-D-21-10726R1 

Molecular and physiological characterization of *Fusarium* strains associated with different diseases in date palm 

Dear Dr. Saleh:

I'm pleased to inform you that your manuscript has been deemed suitable for publication in PLOS ONE. Congratulations! Your manuscript is now with our production department. 

Kind regards, 

on behalf of

Dr. Himanshu Sharma 

Academic Editor

PLOS ONE